

# Prognostic significance of integrating total metabolic tumor volume and EGFR mutation status in patients with lung adenocarcinoma

Maoqing Jiang[1,2], Xiuyu Guo[1], Ping Chen[3], Xiaohui Zhang[1], Qiaoling Gao[1], Jingfeng Zhang[1] and Jianjun Zheng[1]

[1] Department of Radiology, Ningbo No. 2 Hospital, Ningbo, Zhejiang, China
[2] Department of Nuclear Medicine, Ningbo No. 2 Hospital, Ningbo, Zhejiang, China
[3] Department of Nephrology, Ningbo No. 2 Hospital, Ningbo, Zhejiang, China

## ABSTRACT

**Background:** The objective of this study was to investigate the prognostic significance of total metabolic tumor volume (TMTV) derived from baseline $^{18}$F-2-fluoro-2-deoxyglucose ($^{18}$F-FDG) positron emission tomography/computed tomography (PET/CT), in conjunction with epidermal growth factor receptor (EGFR) mutation status, among patients with lung adenocarcinoma (LUAD).

**Methods:** We performed a retrospective analysis on 141 patients with LUAD (74 males, 67 females, median age 67 (range 34–86)) who underwent $^{18}$F-FDG PET/CT and had their EGFR mutation status determined. Optimal cutoff points for TMTV were determined using time-dependent receiver operating characteristic curve analysis. The survival difference was compared using Cox regression analysis and Kaplan–Meier curves.

**Results:** The EGFR mutant patients ($n = 79$, 56.0%) exhibited significantly higher 2-year progression-free survival (PFS) and overall survival (OS) rates compared to those with EGFR wild-type ($n = 62$, 44.0%), with rates of 74.2% *vs* 69.2% ($P = 0.029$) and 86.1% *vs* 67.7% ($P = 0.009$), respectively. The optimal cutoff values of TMTV were 36.42 cm$^3$ for PFS and 37.51 cm$^3$ for OS. Patients with high TMTV exhibited significantly inferior 2-year PFS and OS, with rates of 22.4% and 38.1%, respectively, compared to those with low TMTV, who had rates of 85.8% and 95.0% (both $P < 0.001$). In both the EGFR mutant and wild-type groups, patients exhibiting high TMTV demonstrated significantly inferior 2-year PFS and OS compared to those with low TMTV. In multivariate analysis, EGFR mutation status (hazard ratio, HR, 0.41, 95% confidence interval, CI [0.18–0.94], $P = 0.034$) and TMTV (HR 8.08, 95% CI [2.34–28.0], $P < 0.001$) were independent prognostic factors of OS, whereas TMTV was also an independent prognosticator of PFS (HR 2.59, 95% CI [1.30–5.13], $P = 0.007$).

**Conclusion:** Our study demonstrates that the integration of TMTV on baseline $^{18}$F-FDG PET/CT with EGFR mutation status improves the accuracy of prognostic evaluation for patients with LUAD.

Corresponding author
Jianjun Zheng,
zhengjianjun@ucas.ac.cn

# INTRODUCTION

Lung cancer continues to be a prominent contributor to global cancer-related deaths, showing minimal progress in prognosis despite advancements made in the field of diagnosis and treatment approaches (*Siegel et al., 2023*; *Xia et al., 2022*). Lung adenocarcinoma (LUAD) represents a predominant subtype of non-small cell lung cancer (NSCLC), accounting for approximately 40% of all cases among pulmonary malignancies (*Kleczko et al., 2019*; *Zhang et al., 2022*). Advances in molecular research have resulted in the emergence of promising treatments for advanced NSCLC, such as gefitinib, a targeted agent that effectively inhibits epidermal growth factor receptor (EGFR) tyrosine kinase (*Yi et al., 2023*). Patients with NSCLC harboring EGFR mutations and treated with tyrosine kinase inhibitors (TKIs) achieved significantly prolonged progression-free survival (PFS) and/or overall survival (OS) compared to those receiving conventional chemotherapy (*Zhong et al., 2021*; *Liu et al., 2021*; *Greenhalgh et al., 2021*; *Sperduto et al., 2017*; *Cadranel et al., 2012*). The presence of EGFR mutations has been proposed as a crucial determinant of prognosis in individuals with NSCLC (*Wu et al., 2010*; *Choi et al., 2012*; *Deng et al., 2021*), but the predictive significance of EGFR mutations in NSCLC patients remains controversial (*Zhang et al., 2014*; *Lin et al., 2017*). Due to a paucity of studies specifically investigating the prognostic impact stratified by clinical TNM stages, histologic subtypes, or metabolic phenotypes on $^{18}$F-2-fluoro-2-deoxyglucose ($^{18}$F-FDG) positron emission tomography/computed tomography (PET/CT), it becomes challenging to adequately control for confounding variables.

At present, the utilization of $^{18}$F-FDG PET/CT is extensive in the management of lung cancer and has been recognized for providing prognostic insights through metabolic parameters (*e.g.*, maximum standardized uptake value of primary tumors (pSUV$_{max}$), total metabolic tumor volume (TMTV), and whole-body total lesion glycolysis (TLG$_{WB}$)) derived from PET images (*Pellegrino et al., 2019*; *Mahmoud et al., 2022*; *Monaco et al., 2021*; *Chen et al., 2012*). Essentially, patients diagnosed with lung cancer who exhibit elevated pSUV$_{max}$, TMTV and TLG$_{WB}$ values are indicative of a poor prognosis. The occurrence of EGFR mutations in individuals diagnosed with lung cancer generally results in a positive reaction to targeted treatment, which contributes to prolonged survival. However, several studies have reported inconsistent findings regarding the correlation between EGFR mutation and decreased $^{18}$F-FDG uptake in lung cancer, with some indicating a negative association and others suggesting otherwise (*Shi et al., 2022*; *Hong et al., 2020*; *Ko et al., 2014*). Additionally, certain studies have failed to identify any significant correlations between these factors (*Chung et al., 2014*; *Lee et al., 2015*). The inconsistent findings could be attributed to the limited sample size and potential confounding variables, such as the TNM stage and histological subtype of the tumor.

NSCLC can be categorized into two primary subtypes, LUAD and squamous cell carcinoma (SCC), each exhibiting distinct features. *Wang et al. (2020)* demonstrated that it

is crucial to analyze LUAD and lung SCC separately to obtain precise prognostic information due to significant outcome differences between these two distinct cancer types. In this study, we specifically chose LUAD as the subject and hypothesized that the presence of EGFR mutations could serve as a prognostic indicator for patients with LUAD; however, further stratification by metabolic parameters on $^{18}$F-FDG PET/CT is necessary to refine these results.

Therefore, we conducted a retrospective analysis to investigate the prognostic significance of the TMTV derived from baseline $^{18}$F-FDG PET/CT scans in conjunction with EGFR mutation status among patients diagnosed with LUAD.

## MATERIALS AND METHODS

### Patient selection

We retrospectively analyzed a cohort of 1,104 patients diagnosed with lung cancer who underwent $^{18}$F-FDG PET/CT at Ningbo No. 2 Hospital in China between October 2019 and March 2022. To be eligible for the study, patients were required to meet specific criteria, which included: (i) no prior pretreatment before undergoing $^{18}$F-FDG PET/CT, (ii) a confirmed diagnosis of LUAD through histopathological examination, (iii) determination of EGFR mutation status, and (iv) at least 1 month of follow-up. The study included a cohort of 141 individuals who were diagnosed with LUAD, selected according to the established criteria (Fig. 1). Individuals who had smoked less than 100 cigarettes in their lifetime were categorized as never smokers, while the rest of the participants were considered smokers (Kawaguchi et al., 2010). The research plan obtained approval from the Institutional Review Board of Ningbo No. 2 Hospital, and informed consent was not required (protocol No. YJ-NBEY-KY202108401).

### Technique for PET/CT scanning

The PET/CT scan procedure employed a GE Discovery 710 PET scanner (GE Healthcare, Chicago, IL, USA). Prior to the examination, patients were instructed to observe a fasting period of at least 6 h, and their glucose levels were verified to be below 7.0 mmol/L. A dosage of 5.2–7.4 MBq/kg of $^{18}$F-FDG was administered, followed by a PET/CT scan conducted after a time interval of 45–60 min. The low-dose CT scan parameters were set as follows: an X-ray tube voltage of 140 kV, current of 10 mA, rotation duration of 0.5 s, and collimation width of 40 mm. Following this, a three-dimensional PET scan was conducted from the base of the skull to the upper thigh, with each bed position scanned for 2.5 min. An iterative algorithm reconstruction utilizing CT data was employed to acquire PET, CT, and fused PET/CT images. The Xeleris Workstation (GE Healthcare) was utilized for image analysis in transverse, sagittal, and coronal planes.

### Analysis of PET/CT imaging

The PET and CT images were independently evaluated by two experienced nuclear medicine physicians (MQJ and QLG) with a minimum of 10 years of clinical practice. The SUV$_{max}$ value was utilized to quantify the intensity of $^{18}$F-FDG uptake in the lesion, considering abnormal uptake as metabolic activity surpassing that observed in the

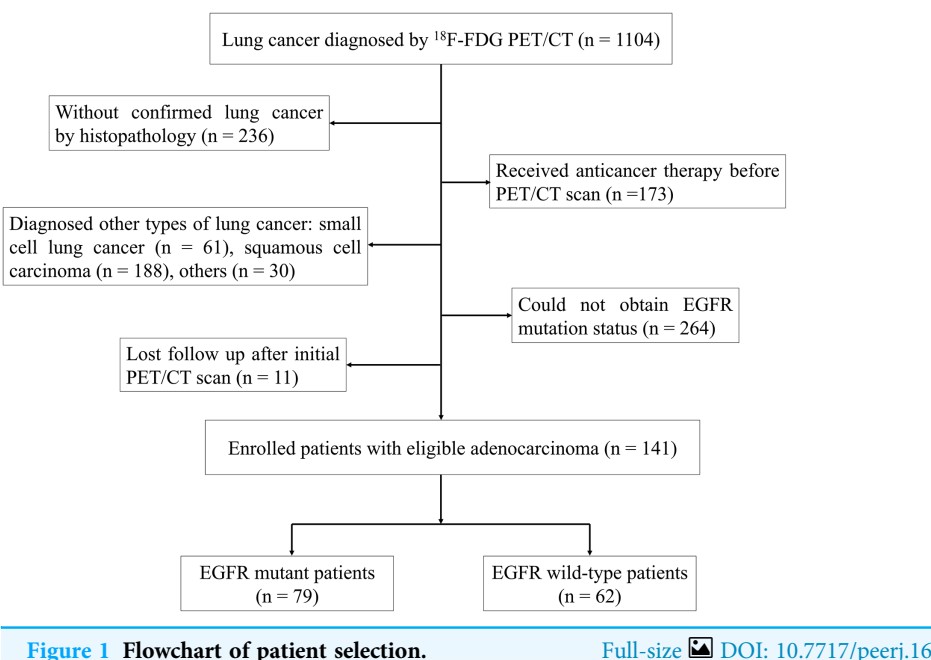

**Figure 1 Flowchart of patient selection.**

surrounding background. A region of interest (ROI) was manually delineated around the tumor lesions, focusing on the area exhibiting the most significant uptake of $^{18}$F-FDG. The $SUV_{max}$ represents the highest standardized uptake value within this ROI. To derive TLG values, a range of margin thresholds were applied to each individual lesion. This involved calculating the product of $SUV_{mean}$ and MTV, which provides an assessment of both tumor burden and metabolic activity. The margin threshold used for determining MTV was equivalent to 41% of the $SUV_{max}$ for each lesion (*Lang et al., 2021*). If the lesions were large and clustered to an extent where individual lesions could not be distinguished, they were classified as a cluster. Specialized software was employed for automated measurements to ensure complete reproducibility. Last, at the patient level, the $TLG_{WB}$ was calculated by summing all lesion values.

## Analysis of EGFR mutations

The presence of EGFR mutations was determined through histological analysis of primary tumors, metastatic lymph nodes or organs obtained *via* surgical resection, fiberoptic bronchoscopy biopsy, or fine-needle aspiration. In all instances, the specimens were fixed in a 10% buffered neutral formalin solution and subsequently embedded in paraffin wax. According to the manufacturer's instructions, DNA was extracted from formalin-fixed paraffin-embedded (FFPE) tissue sections using the QIAamp DNA FFPE Tissue Kit manufactured by Qiagen NV in Venlo, Netherlands. Polymerase chain reaction was carried out on an Mx3000PTM real-time PCR system developed by Stratagene located in La Jolla, CA, USA. The amplification-refractory mutation system, in conjunction with an EGFR 29 Mutation Detection Kit from Amoy Diagnostics (Xiamen, China), was employed to determine the presence of EGFR mutations. Tumors were categorized as harboring

EGFR mutations if exon mutations were detected; otherwise, they were considered wild-type tumors.

## Statistical analysis

The demographic data of the patients are presented using descriptive statistics. Median values with interquartile ranges (IQRs) were reported for metabolic parameters, including $pSUV_{max}$, MTV and TLG of the primary tumors (pMTV and pTLG), TMTV and $TLG_{WB}$. To evaluate variations in continuous variables among different groups, Mann–Whitney tests were conducted. PFS was determined as the time period from the first PET/CT scan until either confirmed disease progression or death, whereas OS was computed from the initial PET/CT scan to either all-cause mortality or last follow-up, whichever occurred earlier. Time-dependent receiver operating characteristic (ROC) curve analysis was employed to determine the optimal cutoff values of $pSUV_{max}$, TMTV and $TLG_{WB}$ for PFS and OS. The predictive performance was assessed by calculating the area under the ROC curve (AUC). The 2-year PFS and OS rates were estimated using Kaplan–Meier curves. Differences in survival between groups were evaluated by the log-rank test. Both univariate and multivariate analyses were conducted using the Cox regression model. R software (version 3.60, http://www.r-project.org) was utilized for statistical analyses, and GraphPad Prism 9.0 (GraphPad Software, San Diego, CA, USA) was used to generate graphs. Statistical significance was determined by a two-tailed $p$ value < 0.05.

## RESULTS

### Patient characteristics

Table 1 presents a comprehensive overview of the clinical and metabolic profiles of patients with wild-type and mutant EGFR, encompassing factors such as age, sex, smoking status, clinical TNM stage, $pSUV_{max}$, TMTV and $TLG_{WB}$ data. In the cohort of 141 participants, comprising 74 males and 67 females, the average age was 66.6 ± 9.8 years (mean ± standard deviation, with a median age of 67 years), ranging from 34 to 86 years. PFS was followed up for a median duration of 16 months (ranging from 1 to 36 months), with IQRs of 9–26 months. Regarding OS, the follow-up period lasted for a median of 21 months (ranging from 3 to 38 months), with IQRs of 11–27 months. During the entire follow-up period, a total of 26 deaths were recorded. The study population as a whole demonstrated a 2-year PFS rate of 66.3% and an OS rate of 78.2%, as shown in Fig. 2.

### Outcomes according to EGFR mutation status

The presence of EGFR mutations was observed in 79 (56.0%) patients, as indicated in Table 1, while the remaining patients (n = 62, 44.0%) were classified as EGFR wild-type. Based on our findings, patients with EGFR mutations demonstrated significantly higher rates of 2-year PFS (Fig. 3A) and OS (Fig. 4A) than those with wild-type EGFR, with rates of 74.2% vs 69.2% (P = 0.029) and 86.1% vs 67.7% (P = 0.009), respectively.
**Table 1 Comparison of clinical features and metabolic parameters between EGFR wild-type and mutant-type patients with adenocarcinoma.**

| Characteristics | Total | EGFR | | P value |
|---|---|---|---|---|
| | | Wild-type | Mutant-type | |
| Age, years | | | | 0.439 |
| Median | 67 | 68 | 67 | |
| Range | 34–86 | 34–86 | 36–85 | |
| Gender (n, %) | | | | 0.006 |
| Male | 74 (52.5) | 41 (29.1) | 33 (23.4) | |
| Female | 67 (47.5) | 21 (14.9) | 46 (32.6) | |
| Smoking history (n, %) | | | | 0.001 |
| Never-smoker | 96 (68.1) | 33 (23.4) | 63 (44.7) | |
| Ever-smoker | 45 (31.9) | 29 (20.6) | 16 (11.3) | |
| Clinical TNM stage (n, %) | | | | 0.570 |
| I | 52 (36.9) | 20 (14.2) | 32 (22.7) | |
| II | 21 (14.9) | 8 (5.7) | 13 (9.2) | |
| III | 19 (13.5) | 10 (7.1) | 9 (6.4) | |
| IV | 49 (34.7) | 24 (17.0) | 25 (17.7) | |
| $pSUV_{max}$ | | | | 0.044 |
| Median | 8.96 | 10.53 | 8.56 | |
| IQR | 6.31–12.91 | 7.23–14.64 | 5.91–12.00 | |
| pMTV | | | | 0.057 |
| Median | 6.87 | 9.95 | 4.39 | |
| IQR | 2.59–229.39 | 3.25–38.0 | 2.52–19.95 | |
| pTLG | | | | 0.032 |
| Median | 36.93 | 62.98 | 20.65 | |
| IQR | 9.77–188.5 | 12.08–249.3 | 9.05–116.1 | |
| TMTV | | | | 0.048 |
| Median | 14.83 | 28.43 | 9.47 | |
| IQR | 3.72–54.86 | 4.87–74.39 | 3.52–45.42 | |
| TLG | | | | 0.025 |
| Median | 88.17 | 147.4 | 30.73 | |
| IQR | 12.67–322.9 | 25.52–508.1 | 10.43–249.9 | |

## Outcomes according to metabolic parameters

In our study, we analyzed various metabolic parameters, including $pSUV_{max}$, pMTV, pTLG, TMTV and $TLG_{WB}$. The AUCs for predicting 2-year PFS were 0.791, 0.804, 0.842, 0.898 and 0.888, respectively, while those for predicting 2-year OS were 0.809, 0.822, 0.855, 0.911 and 0.894, respectively. The parameter with the highest predictive value in both PFS and OS was TMTV; therefore, it was chosen to evaluate prognostic significance in our studies. The optimal cutoff values for TMTV were determined to be 36.42 and 37.51 $cm^3$ for predicting 2-year PFS and OS, respectively. Patients with high TMTV exhibited significantly inferior 2-year PFS (Fig. 3B, $P < 0.001$) and OS (Fig. 4B, $P < 0.001$), with rates

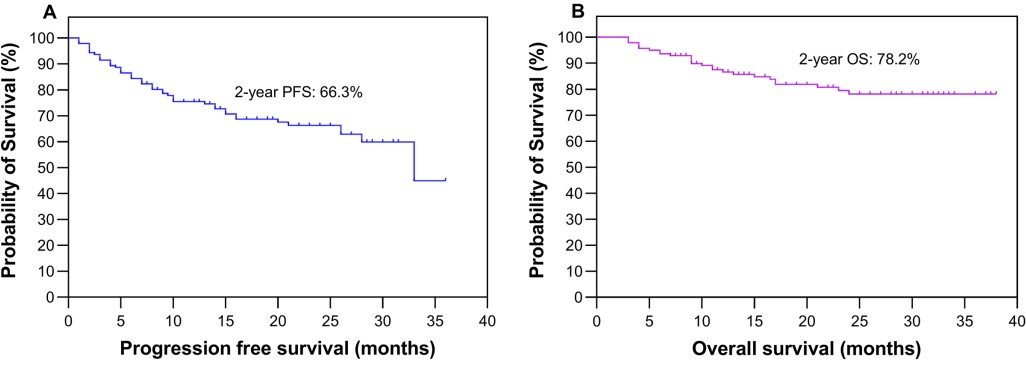

**Figure 2 Survival analysis.** Kaplan–Meier plots depicting the progression-free survival (A) and overall survival (B) of all patients.

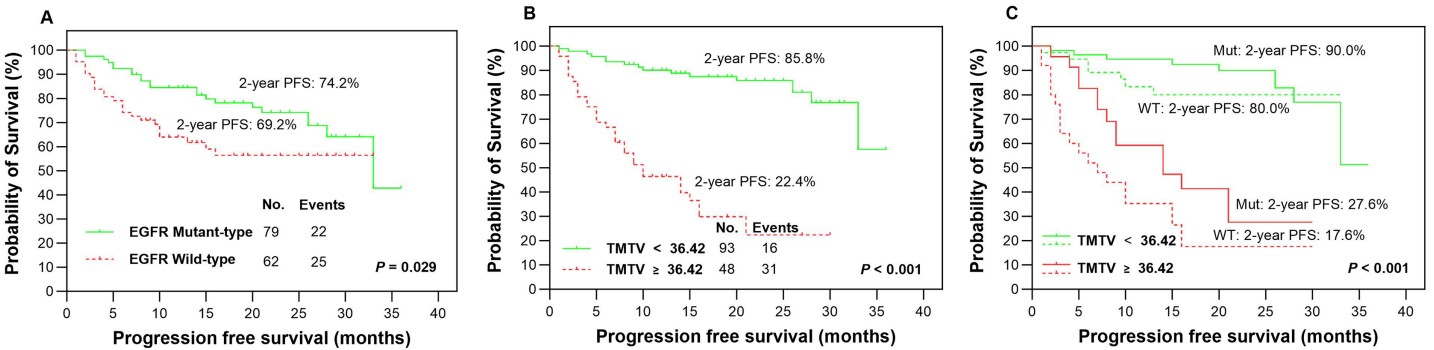

**Figure 3 Survival comparision.** Kaplan-Meier plots were generated to analyze progression-free survival (PFS) in patients based on their EGFR mutation status (mutant-type *vs* wild-type, A) and TMTV (≥36.42 cm³ *vs* <36.42 cm³, B). The optimal cutoff value of TMTV was used to rede-termine PFS in patients with LUAD, specifically in the EGFR mutant group (solid line, C) and wild-type group (dotted line, C). Mut, mutant type; WT, wild-type.

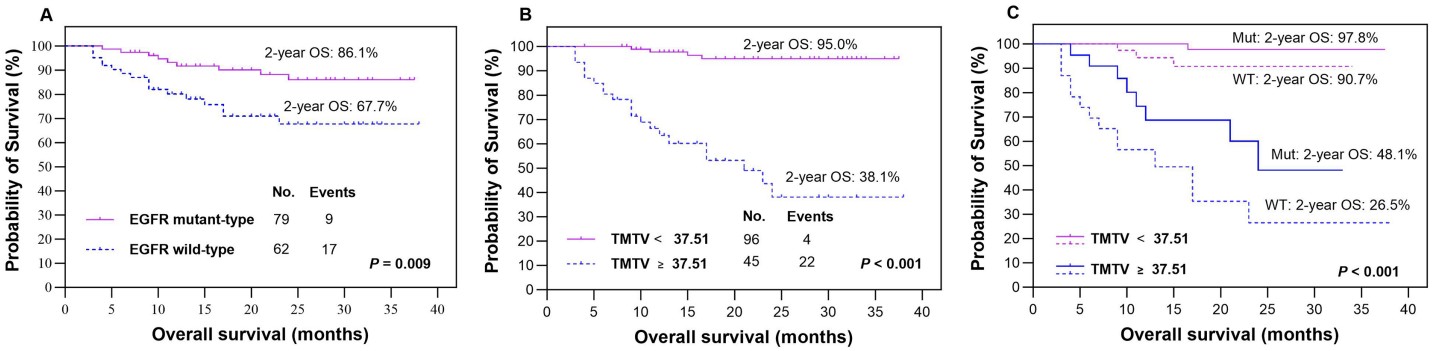

**Figure 4 Survival comparision.** Kaplan–Meier plots were generated to analyze overall survival (OS) in patients based on their EGFR mutation status (mutant-type *vs* wild-type, A) and TMTV (≥37.51 cm³ *vs* <37.51 cm³, B). The optimal cutoff value of TMTV was used to re-stratify OS in patients with LUAD, specifically in the EGFR mutant group (dotted line, C) and wild-type group (dotted line, C). Mut, mutant type; WT, wild-type.

**Table 2 Distribution of patient characteristics between TMTV groups in assessing progression free survival and overall survival.**

| Characteristics | PFS | | | OS | | |
|---|---|---|---|---|---|---|
| | TMTV ≥ 36.42 cm³ | TMTV < 36.42 cm³ | *P* value | TMTV ≥ 37.51 cm³ | TLG < 37.51 cm³ | *P* value |
| Age (y, Median, IQR) | 68 (61–74) | 67 (62–72) | 0.769 | 67 (61–74) | 67 (62–72) | 0.889 |
| Gender | | | 0.028 | | | 0.018 |
| Male | 42 (62.7) | 32 (43.2) | | 45 (62.5) | 29 (42.0) | |
| Female | 25 (37.3) | 42 (56.8) | | 27 (37.5) | 40 (58.0) | |
| Smoking status | | | 0.106 | | | 0.074 |
| Never | 41 (61.2) | 55 (74.3) | | 44 (61.1) | 52 (75.4) | |
| Ever or current | 26 (38.8) | 19 (25.7) | | 28 (38.9) | 17 (24.6) | |
| TNM stage | | | <0.001 | | | <0.001 |
| I–II | 9 (13.4) | 64 (86.5) | | 11 (15.3) | 62 (89.9) | |
| III–IV | 58 (86.6) | 10 (13.5) | | 61 (84.7) | 7 (10.1) | |
| EGFR mutation status | | | 0.065 | | | 0.007 |
| Mutant-type | 32 (47.8) | 47 (63.5) | | 32 (44.4) | 47 (68.1) | |
| Wild-type | 35 (52.2) | 27 (36.5) | | 40 (55.6) | 22 (31.9) | |

of 22.4% and 38.1%, respectively, compared to those with low TMTV, who had rates of 85.8% and 95.0%.

## Outcomes according to the integration of EGFR mutation status and TMTV

In the EGFR mutant group, patients with a high TMTV (≥36.42 cm³) exhibited significantly inferior 2-year PFS compared to those with a low TMTV (<36.42 cm³), while similar observations were made in the EGFR wild-type group, where the respective 2-year PFS rates were 27.6% *vs* 90.0% (Fig. 3C, *P* < 0.001) and 17.6% *vs* 80.0% (Fig. 3C, *P* < 0.001). Patients with EGFR mutations and a high TMTV of ≥37.51 cm³ exhibited significantly lower 2-year OS than those with a low TMTV of <37.51 cm³. The same trend was seen in the EGFR wild-type group, where the respective 2-year OS rates were 48.1% *vs* 97.8% (Fig. 4C, *P* < 0.001) and 26.5% *vs* 90.7% (Fig. 4C, *P* < 0.001).

## Clinical distribution based on TMTV in predicting PFS and OS

Furthermore, we conducted an assessment of the clinical characteristics and EGFR mutation status in relation to both PFS and OS, stratifying patients based on the optimal cutoff values with low and high TMTV (Table 2). Patients with high TMTV often presented in advanced stage and were predominantly male, while there was no significant difference in age or smoking status. The presence of EGFR mutations, typically observed in patients with low TMTV, demonstrated a significant difference in OS but not in PFS (Table 2).

**Table 3 Univariate and multivariate Cox regression analysis of clinical factors, metabolic parameters and EGFR mutation status in relation to patient's outcome.**

| Characteristic | Univariate analysis for PFS | | | Multivariate analysis for PFS | | | Univariate analysis for OS | | | Multivariate analysis for OS | | |
|---|---|---|---|---|---|---|---|---|---|---|---|---|
| | HR | 95% CI | *P*-value | HR | 95% CI | *P*-value | HR | 95% CI | *P*-value | HR | 95% CI | *P*-value |
| Age | 1.03 | [1.00–1.06] | 0.080 | | | | 1.05 | [1.01–1.10] | 0.023 | 1.04 | [1.00–1.08] | 0.041 |
| Gender | | | 0.150 | | | | | | 0.141 | | | |
| Male | Reference | | | | | | Reference | | | | | |
| Female | 0.65 | [0.36–1.17] | 0.150 | | | | 0.55 | [0.24–1.22] | 0.141 | | | |
| Smoking history | | | 0.049 | | | 0.085 | | | 0.179 | | | |
| Never-smokers | Reference | | | | | | Reference | | | | | |
| Smokers | 1.79 | [1.00–3.19] | 0.049 | 1.72 | [0.93–3.18] | 0.085 | 1.70 | [0.78–3.71] | 0.179 | | | |
| TNM stage | | | <0.001 | | | <0.001 | | | 0.008 | | | |
| I | Reference | | | Reference | | | Reference | | | Reference | | |
| II | 1.76 | [0.29–10.6] | 0.535 | 1.64 | [0.27–10.1] | 0.594 | 5.12 | [0.46–56.4] | 0.183 | 1.95 | [0.15–24.6] | 0.606 |
| III | 14.5 | [4.02–52.0] | <0.001 | 8.53 | [2.18–33.4] | 0.002 | 21.3 | [2.56–177] | 0.005 | 5.55 | [0.51–59.9] | 0.158 |
| IV | 18.3 | [5.57–60.2] | <0.001 | 11.7 | [3.23–42.6] | <0.001 | 25.2 | [3.34–189] | 0.002 | 4.43 | [0.43–45.1] | 0.209 |
| TMTV | | | <0.001 | | | 0.007 | | | <0.001 | | | <0.001 |
| <36.42 | Reference | | | Reference | | | / | | | | | |
| ≥36.42 | 7.34 | [3.88–13.9] | <0.001 | 2.59 | [1.30–5.13] | 0.007 | | | | | | |
| <37.51 | / | | | | | | Reference | | | Reference | | |
| ≥37.51 | | | | | | | 17.5 | [5.99–51.3] | <0.001 | 8.08 | [2.34–28.0] | <0.001 |
| EGFR mutation | | | 0.032 | | | 0.078 | | | 0.012 | | | 0.034 |
| Wild-type | Reference | | | Reference | | | Reference | | | Reference | | |
| Mutant-type | 0.53 | [0.30–0.95] | 0.032 | 0.59 | [0.32–1.06] | 0.078 | 0.35 | [0.16–0.79] | 0.012 | 0.41 | [0.18–0.94] | 0.034 |

**Note:**
HR, hazard ratio; CI, confidence interval; OS, overall survival.

## Univariate and multivariate Cox regression analysis of survival

In the univariate analysis, smoking status, TNM stage, TMTV and EGFR mutation status were identified as significant predictors of PFS in patients with LUAD. Additionally, TNM stage, TMTV and EGFR mutation status were found to be predictive of OS (Table 3). The significant factors were subjected to multivariate analysis, revealing that TNM stage and TMTV independently predicted PFS, while EGFR mutation status and TMTV independently predicted OS (Table 3).

## DISCUSSION

In the present study, we have demonstrated that both the EGFR mutation status and the TMTV, determined on baseline [18]F-FDG PET/CT, are independent prognostic factors for OS in patients with LUAD. Furthermore, we found that TMTV is also an independent prognostic factor for PFS in LUAD patients. When evaluating the prognostic significance of EGFR mutation status, it is crucial to consider the level of TMTV. The combination of pretreatment TMTV and EGFR mutation status has the potential to enhance accuracy in predicting prognosis and aid in decision-making regarding intensive therapy.

The prognostic role of EGFR mutation status in patients with lung cancer was investigated as early as 2004 to 2005 (*Taron et al., 2005*; *Lynch et al., 2004*). Patients with EGFR mutations tend to have a high response to TKIs, leading to prolonged survival (*Han et al., 2005*). Nonetheless, there has been considerable fluctuation in the outcomes over the past 20 years. *Mitsudomi et al. (2005)* demonstrated that the presence of genetic alterations in the EGFR gene is associated with improved survival outcomes following gefitinib therapy among NSCLC patients who experience recurrence after surgery. However, *Deng et al. (2021)* discovered a contradictory result indicating that EGFR was a significant negative prognostic indicator in patients with radiologic solid and different forms of LUAD. In comparison to the wild-type group, patients with EGFR mutations exhibited notably higher occurrences of brain and bone metastases (*Deng et al., 2021*). Interestingly, *Liu et al. (2014)* and *Li et al. (2019)* found that primary resected LUAD does not exhibit substantial prognostic significance in relation to EGFR mutations.

Overall, several factors may have contributed to these disparate findings. First, the patient cohorts exhibited heterogeneity in terms of TNM staging, with some only at stage I and having undergone radical surgery while others were at stages I–IV with varying treatment modalities (*Deng et al., 2021*; *Liu et al., 2014*; *Li et al., 2019*). Second, the sample size enrolled in these studies varied greatly from 59 to 1,512 patients (*Deng et al., 2021*; *Mitsudomi et al., 2005*). Third, there was also histological diversity among the cohorts, with some being enrolled as NSCLC and others solely as LUAD (*Zhang et al., 2014*; *Li et al., 2019*). To a certain extent, the prognostic significance of EGFR mutation status in lung cancer has captured the attention of researchers. Our study demonstrates that EGFR mutation is an independent and favorable prognostic factor for patients with LUAD.

The incidence of EGFR mutations is reportedly higher in patients diagnosed with LUAD, particularly among female individuals, never-smokers, and East Asian populations (*Shi et al., 2015*; *Shi et al., 2014*). Our study consistently observed these findings. [18]F-FDG PET/CT has become a widely utilized tool in the management of lung cancer, encompassing diagnosis, treatment response assessment and prognostication (*Lim et al., 2022*; *Peng et al., 2022*). In terms of prognosis, high metabolic activity as measured by [18]F-FDG PET/CT is typically indicative of poor survival outcomes in patients with LUAD. Relevant investigations have been conducted to explore the associations between EGFR mutation status and metabolic parameters on FDG PET/CT (*Jiang et al., 2022*; *Jiang et al., 2023*; *Guo et al., 2021*). We previously observed that male patients with NSCLC harboring EGFR mutations frequently exhibit low $pSUV_{max}$ (*Jiang et al., 2023*; *Zhang et al., 2023*). Similar findings were also reported by *Wang et al. (2022)*. The role of EGFR signaling is pivotal in facilitating aerobic glycolysis in LAD cells harboring EGFR mutations (*Makinoshima et al., 2014*). Decreased expression of glucose transporter 3 (GLUT3), responsible for glucose transport, was observed in TKI-responsive LAD cells (*Makinoshima et al., 2014*), which may contribute to the lower metabolic activity associated with the EGFR mutant type. In this study, a lower level of metabolic parameters, including $pSUV_{max}$, pTLG, TMTV and $TLG_{WB}$, was associated with a higher incidence of EGFR mutations. However, a number of factors may influence the correlations between EGFR mutation status and $SUV_{max}$ in LUAD, particularly with regard to smoking status

(*Gao et al., 2023*). Furthermore, a greater intratumor heterogeneity factor was observed in EGFR-mutant LUAD patients than in those with wild-type EGFR (*Ni et al., 2023*). Therefore, the integration of EGFR mutation status and $^{18}$F-FDG metabolic activity is imperative and holds paramount significance for a comprehensive evaluation of prognostic outcomes in patients with LUAD.

Accordingly, we investigated the prognostic value of various metabolic parameters, including $pSUV_{max}$, pMTV, pTLG, TMTV and $TLG_{WB}$. Upon comparison of these parameters, TMTV exhibited the highest prognostic efficacy for patients diagnosed with LUAD. A meta-analysis comprising thirty-six studies and 5,807 patients demonstrated that elevated $pSUV_{max}$, MTV, and $TLG_{WB}$ were associated with a poor prognosis in surgical NSCLC patients (*Liu et al., 2016*). *Salavati et al. (2017)* found that volumetric parameters derived from both primary tumors and whole-body lesions exhibit comparable prognostic value for survival in stage IIB/III NSCLC patients. As stated, both the EGFR mutation status and metabolic parameters can serve as crucial factors for assessing treatment response sensitivity and prognosis, exhibiting a significant correlation between them. However, there have been limited studies integrating EGFR mutation status and metabolic parameters to evaluate the prognosis of patients with LUAD. In our findings, not only in EGFR wild-type patients but also in those with EGFR mutations, the parameter of TMTV could effectively stratify them into distinct prognostic groups. It is crucial to take into account the volumetric parameter of TMTV when prognosticating based on EGFR mutation status.

However, it is important to acknowledge its limitations. First, the delineation of lesions relied on a single threshold technique, and although CT images were incorporated for improved accuracy, the choice of threshold value can still impact the quantification of tumor volume, average SUV, and TMTV. Future research should explore alternative thresholds to optimize these measurements since there is currently no standardized approach for determining the optimal cutoff value for $SUV_{max}$. Second, the duration of follow-up was relatively short, and the sample size was limited. Third, this study was conducted retrospectively at a single center. Therefore, it is necessary to validate these findings through larger-scale prospective randomized studies involving multiple institutions.

## CONCLUSIONS

In conclusion, both the EGFR mutation status and the TMTV measured on baseline $^{18}$F-FDG PET/CT can independently serve as prognostic factors for OS in patients with LUAD. Furthermore, the TMTV is also an independent predictor for PFS in LUAD patients. Integrating them may enhance the predictive accuracy for patient outcomes, which could be valuable for clinicians when making decisions regarding treatment modalities and follow-up.

### Funding

This work was supported by the Exploration Project of Natural Science Foundation of Zhejiang Province (grant No. LTGY23H180004), the Ningbo Youth Science and Technology Innovation Leading Talent Project (No. 2023QL057), the Zhu Xiu Shan Talent Project of Ningbo No. 2 Hospital, Ningbo, China (Grant Nos. 2023HMJQ29 and 2021HMYQ07), the Ningbo Clinical Research Center for Medicine Imaging (grant No. 2021L003), and the Provincial and Municipal Co-construction Key Discipline for Medical Imaging (grant No. 2022-S02). The funders had no role in study design, data collection and analysis, decision to publish, or preparation of the manuscript.

### Grant Disclosures

The following grant information was disclosed by the authors:
Exploration Project of Natural Science Foundation of Zhejiang Province: LTGY23H180004.
Ningbo Youth Science and Technology Innovation Leading Talent Project: 2023QL057.
Zhu Xiu Shan Talent Project: 2023HMJQ29 and 2021HMYQ07.
Ningbo Clinical Research Center for Medicine Imaging: 2021L003.
Provincial and Municipal Co-construction Key Discipline for Medical Imaging: 2022-S02.

### Competing Interests

The authors declare that they have no competing interests.

### Author Contributions

- Maoqing Jiang conceived and designed the experiments, performed the experiments, analyzed the data, prepared figures and/or tables, authored or reviewed drafts of the article, and approved the final draft.
- Xiuyu Guo performed the experiments, analyzed the data, prepared figures and/or tables, and approved the final draft.
- Ping Chen performed the experiments, analyzed the data, prepared figures and/or tables, authored or reviewed drafts of the article, and approved the final draft.
- Xiaohui Zhang performed the experiments, analyzed the data, prepared figures and/or tables, and approved the final draft.
- Qiaoling Gao performed the experiments, prepared figures and/or tables, and approved the final draft.
- Jingfeng Zhang performed the experiments, authored or reviewed drafts of the article, and approved the final draft.
- Jianjun Zheng conceived and designed the experiments, performed the experiments, authored or reviewed drafts of the article, and approved the final draft.

### Human Ethics

The following information was supplied relating to ethical approvals (*i.e.*, approving body and any reference numbers):

The study was approved by the Institutional Review Board of Ningbo No. 2 Hospital (protocol No. YJ-NBEY-KY202108401).

## Data Availability

The raw measurements are available in the Supplemental File.

## Supplemental Information

Supplemental information for this article can be found online at http://dx.doi.org/10.7717/peerj.16807#supplemental-information.

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
