# Peer review of "Prognostic significance of integrating total metabolic tumor volume and EGFR mutation status in patients with lung adenocarcinoma"

_PeerJ, doi:10.7717/peerj.16807_

## Round 0.1 · original submission · Minor Revisions

Dear Dr. Jiang,

Thank you for your submission to PeerJ. We have received reviewer comments for your manuscript entitled "Prognostic significance of integrating total metabolic tumor volume and EGFR mutation status in patients with lung adenocarcinoma" After careful consideration, we have decided to invite minor revision of the manuscript.

Please address all the reviewers' comments point-by-point and highlight them in the revised version.


With kind regards,
Abhishek Tyagi
Academic Editor
PeerJ Life & Environment

Reviewer 1 ·

Basic reporting

The manuscript is easily understandable. However, the manuscript contains no information regarding treatments for lung cancer which is important to determine survival outcomes.
Other minor points:
1. Line 46, "showing minimal progress in progress". I disagree with this statement. In the past few years, the use of TKIs and immune checkpoint inhibitors have greatly improved survival of patients with lung adeno.
2. Line 49, the citations "Kleczko 2019; Zhang 2022" do not support the authors' claim that lung adeno is common in NSCLC.
3. Line 100, "Table 1 presents a comprehensive..." this should be in the result section not the method section.
4. Line 177, "the average age was 66.6+/-9.8 years" is unclear. I guess this should be mean and SD?
5. Table 2: I don't quite understand what additional information that Table 2 presents, especially with the separation of PFS and OS. A better way to demonstrate this might be an univariable analysis of factors that might be related to high and low TMTV.
6. For patients with EGFR mutation, any of them had exon 20 mutation? Patients with exon 20 mutation have poor response to traditional TKIs.

Experimental design

There is a major flaw in the study design. It included patients with all stages of lung adeno --- from localized disease to metastatic disease. Patients with higher TMTV have higher disease burden, who are more likely to have metastatic disease and therefore worse survival outcomes. Therefore, TMTV is just a surrogate for lung cancer staging, which is the best known factor of affecting cancer outcomes. I suggest that the authors only focus on patients with metastatic lung cancer (i.e. stage IV lung cancer) for a more homogeneous cohort.

Validity of the findings

The author concluded that the integration of TMTV and EGFR mutation status improves accuracy of predicting outcomes. However, this is not supported by the results presented in the manuscript. The authors showed that patients with EGFR mutation had better PFS/OS than those without EGFR mutation, which is well known, and patients with higher TMTV had better PFS/OS than those with lower TMTV, which is also evident because patients with higher TMTV have later stages of lung cancer. It is unclear how integrating the two would improve prediction of outcomes. One way to improve this is to focus only on stage IV lung adeno, and demonstrate K-M curves of PFS/OS of EGFR mut, EGFR mut+high TMTV, EGFR mut+low TMTV in a single figure. If integration of TMTV to EGFR mutation status could improve prediction value, we would see these three curves separate.

Additional comments

None

Reviewer 2 ·

Basic reporting

This paper investigates the prognostic significance of total metabolic tumor volume (TMTV) in conjunction with epidermal growth factor receptor (EGFR) mutation status among patients with lung adenocarcinoma (LUAD). This work appears to be well written and structure.

Experimental design

Using a retrospective analysis, the authors were able to determine that integration of TMTV with EGFR status improves the accuracy of prognostic evaluation.

Validity of the findings

This is a well-done study. To improve the accuracy of prognostic evaluation for patients with LUAD is pivotal. I do not have any major criticisms for this paper.

·

Basic reporting

The manuscript under review presents a robust and well-crafted document with an innovative perspective on advancing new predictors in lung carcinoma. The writing is succinct and clear, featuring a well-supported literature introduction and solid supplementary materials.

Introduction Enhancement:
1. Strengthen the introduction by delving further into the predominant mutations in LUAD. Discuss their potential impact on specific metabolic or other signaling pathways (lines 53-59). Refer to the provided annotated comments in the manuscript for guidance.

Experimental design

The experimental design is commendable, aligning seamlessly with the journal's Aims and Scope. The research question is clearly defined (line 84), and the presented data effectively addresses the author's hypothesis, demonstrating significant predictive benefits that are both relevant and meaningful. The methods are well-described and pertinent to the presented data. Refer to the manuscript annotated text for specific comments.

Validity of the findings

The conclusions are well-articulated, directly tied to the original research question, and effectively supported by relevant literature.
2. Consider consistent formatting for figures, either using the full "figure" or the short "Fig" spelling, but not both (lines 182, 188). See comments in the annotated manuscript text for further clarification.
3. Propose combining Figure 3,4 parts C+D to enhance visualization of the disparities between WT/Mut EGFR and high/low TMTV status. Refer to the annotated manuscript text for detailed suggestions.
4. Offer a plausible explanation for why Mut status may impact TMTV. Expand on mutation mechanisms if literature is available (Line 275). Consult the comments in the annotated manuscript text for guidance.

Additional comments

In conclusion, the manuscript displays a strong foundation but can benefit from the outlined improvements to enhance clarity, consistency, and visual representation.

---

## Round 0.2 · accepted · Accept

Dear Dr. Jiang,

Thank you for your submission to PeerJ.

I am writing to inform you that your manuscript - Prognostic significance of integrating total metabolic tumor volume and EGFR mutation status in patients with lung adenocarcinoma - has been Accepted for publication.

Congratulations, and thank you for your submission.

With kind regards,

Abhishek Tyagi
Academic Editor
PeerJ Life & Environment

Reviewer 1 ·

Basic reporting

The authors have addressed the comments. I would recommend the authors to acknowledge the limitation that the study population was heterogeneous, including patients with various lung cancer stages from stage I to stage IV.

Experimental design

none

Validity of the findings

none

Additional comments

none

·

Basic reporting

All previous review comments and suggestions were well incorporated, making the improved manuscript robust and well-crafted document.

Experimental design

All previous review comments and suggestions were well incorporated, making the improved manuscript robust and well-crafted document.

Validity of the findings

All previous review comments and suggestions were well incorporated, making the improved manuscript robust and well-crafted document.

Additional comments

All previous review comments and suggestions were well incorporated, making the improved manuscript robust and well-crafted document.